



# Ambient Nitro-Aromatic Compounds - Biomass Burning versus Secondary Formation in rural China

**Christian Mark Garcia Salvador**[1], Rongzhi Tang[2], Michael Priestley[1], Lin Jie Li[1], Epameinondas Tsiligiannis[1, #], Michael Le Breton[1, #], Wenfei Zhu[3], Limin Zeng[1], Hui Wang[1], Ying Yu[1], Min Hu[1], Song Guo[2, 4*], Mattias Hallquist[1,*]

[1]*Department of Chemistry and Molecular Biology, University of Gothenburg, Gothenburg, Sweden*
[2]*College of Environmental Sciences and Engineering, Peking University, Beijing, China*
[3]*Shanghai Academy of Environmental Sciences, Shanghai 200233, China*
[4]*International Joint Laboratory for Regional Pollution Control, Ministry of Education, Beijing, 100871, China P. R*
[#]*Now at Volvo Group Trucks and Technology Method and Technical Development, Gothenburg, Sweden*

*Corresponding authors: Mattias Hallquist (hallq@chem.gu.se) and Song Guo (songguo@pku.edu.cn)*

## ABSTRACT

Nitro-aromatic compounds (NACs) were measured hourly at a rural site in China during wintertime to monitor the changes due to local and regional impacts of biomass burning (BB). Concurrent and continuous measurements of the concentrations of 16 NACs in the gas and particle phases were performed with a time-of-flight chemical ionization mass spectrometer (CIMS) equipped with a Filter Inlet for Gas and Aerosol (FIGAERO) unit using iodide as the reagent ion. NACs accounted for <2% of the mass concentration of organic matter (OM) and total particulate matter (PM), but the total particle mass concentrations of these compounds can reach as high as 1000 ng m$^{-3}$ (299 ng m$^{-3}$ ave.), suggesting that they may contribute significantly to the radiative forcing effects of atmospheric particles. Levels of gas-phase NACs were highest during the daytime (15:00-16:00 local time, L.T.), with a smaller night-time peak around 20:00 L.T. Box-model simulations showed that this occurred because the rate of NAC production from gas-phase sources exceeded the rate of loss, which occurred mainly via the OH reaction and to a lesser degree via photolysis. Data gathered during extended periods with high contributions from primary BB sources (resulting in 40-60% increases in NAC concentrations) were used to characterize individual NACs with respect to gas-particle partitioning and the contributions of regional secondary processes (i.e. photochemical smog). On days without extensive BB, secondary formation was the dominant source of NACs and NAC levels correlated strongly with the ambient ozone concentration. Analyses of individual NACs in the regionally aged plumes sampled on these days allowed precursors such as phenol and catechol to be linked to their





NAC derivatives (i.e. nitrophenol and nitrocatechol). Correlation analysis using the high time resolution data and box-model simulation results constrained the relationships between these compounds and demonstrated the contribution of secondary formation processes. Furthermore, 13 of 16 NACS were classified according to primary or secondary formation process. Primary emission was

the dominant source (accounting for 60-70% of the measured concentrations) of 5 of the 16 studied NACs, but secondary formation was also a significant source. Photochemical smog thus has important effects on brown carbon levels even during wintertime periods dominated by primary air pollution in rural China.

## 1.    Introduction

Nitro-aromatic compounds (NACs) are aromatic compounds containing at least one nitro ($-NO_2$) functional group attached directly to a benzene ring. They are considered anthropogenic compounds; there is little or no evidence of their formation from natural sources. The atmospheric production and behavior of NACs has attracted interest due to their role in the formation of brown carbon (BrC) aerosols (Xie et al., 2017;Xie et al., 2019;Lin et al., 2016;Mohr et al., 2013). Nitro-

aromatic compounds such as nitrophenol (NP), nitrocatechol (NC), and dinitrophenol (DNP) absorb light in the near-ultraviolet and visible regions, which can cause positive radiative forcing (Zhang et al., 2017). While NACs may constitute only a minor fraction of aerosols (<2%) (Mohr et al., 2013;Wang et al., 2017), they can account for as much as 50% of the light absorption coefficient of SOA at 365 nm (Xie et al., 2017). In addition to their roles as potential climate forcers, NACs can adversely affect

human health: exposure to nitrophenol can cause blood disorders that retard the delivery of oxygen to tissues and organs ((ATSDR), 2015). The harmful effects of NACs and their strong effects on radiation balances even at minute concentrations necessitate a robust understanding of their atmospheric behavior and sources.

The origins of atmospheric NACs can vary depending on the prevailing atmospheric conditions

and local levels of precursors and oxidants. Traffic and biomass burning (BB) are the main sources of NACs (Inomata et al., 2016;Perrone et al., 2014;Le Breton et al., 2019). BB is considered a major driver of atmospheric NAC formation (Kahnt et al., 2013;Laskin et al., 2015) because the combustion of coal and wood, which leads to  causes thermal degradation and pyrolysis of lignins, results to strong emission





of substituted phenols including 1,2-benzenediols (catechols) and cresol/methylphenols that are precursors for the formation of NACs. Accordingly, several lab and field studies have demonstrated the formation of NACs originating from wood-burning under various atmospheric conditions. For instance, molecular characterization of aged BB plumes from a nationwide bonfire festival revealed a suite of nitro-aromatic compounds that accounted for 50–80% of the total visible light absorption (Lin et al., 2017). Additionally, combustion of pine under high flaming conditions resulted in the emission of 8.1 mg/kg of NACs, primarily NC and NP (Hoffmann et al., 2007). Fourteen NAC species were detected in laboratory simulations of open BB, with nitrocatechol having the highest mass concentration regardless of the choice of wood fuel (Xie et al., 2019). The same study reported a significant contribution of NACs to absorption at 365 nm ($Abs_{365}$), indicating that NACs are strong Brown Carbon (BrC) chromophores. Cloud water samples collected during a period of extensive wheat straw burning at a mountain site in Northern China also showed the presence of NP, NC, and their derivatives (Desyaterik et al., 2013).

The atmospheric abundance of NACs can also be attributed to secondary oxidation of precursor aromatic compounds (Yuan et al., 2016). The photooxidation and subsequent nitration of benzene and toluene yields NP and methylnitrophenol (MNP), respectively. Oxidation of precursors is initiated by hydroxyl (OH) and nitrate ($NO_3$) radicals, which can cause both daytime and nighttime formation of NACs. Wood burning processes also emit significant quantities of aromatic compounds with OH-substituents such as phenol and catechol, which can be transformed into NP and NC under high $NO_x$ conditions (Finewax et al., 2018). In the same way, oxidation of mononitrates generates nitrophenoxy radicals and similar compounds that produce DNP and other dinitrates. In most of these processes, the ambient concentration of $NO_2$ is the main determinant of the rate of NAC formation until the system/site reaches a $NO_x$-saturated regime under which further increases in $NO_2$ levels do not increase NAC formation (Yuan et al., 2016; Wang et al., 2019). In addition to the typical gas-phase photooxidation mechanism, NAC formation can occur via aqueous-phase oxidation, which is favored by a high atmospheric liquid content and/or the presence of clouds that provide aqueous surfaces on which oxidation reactions can proceed (Lüttke et al., 1997; Vione et al., 2005). In China, it was calculated that



domestic and open burning were responsible for the emission of 640 tons of NACs (Wang et al., 2017). However, there is considerable uncertainty in this estimate because the amount of material burned varies from year to year. Consequently, there is a clear need for further studies on the emissions of nitro-aromatic compounds in China.

NACs are typically collected using aerosol filters over an extended period (e.g. 12 hours to 1 day). Particle-bound NACs can then be extracted and analyzed using Gas Chromatography (GC) followed by Mass Spectrometry (MS) (Cecinato et al., 2005) or Ultra/High-Pressure Liquid Chromatography (UP/HPLC) (Wang et al., 2018;Wang et al., 2019;Iinuma et al., 2010;Finewax et al., 2018;Kahnt et al., 2013;Xie et al., 2019). Recent advances in measurement techniques have enabled

fast and semi-continuous measurement of NACs using high-resolution Time-of-Flight Chemical Ionization Mass Spectrometry (HR-ToF-CIMS) (Le Breton et al., 2019;Mohr et al., 2013;Yuan et al., 2016;Gaston et al., 2016). The high resolving power of CIMS (m/$\Delta$m > 3000) has facilitated the detection of analytes with very low detection limits. Additionally, by using a Filter Inlet for Gases and AEROsols (FIGAERO) unit, the concentrations of NACs and other species in both the gas and aerosol

phases can be analyzed simultaneously without need for sample preparation (e.g. by solvent extraction) (Lopez-Hilfiker et al., 2014;Gaston et al., 2016).

In this work, the formation of gas- and particle-phase NACs in a rural area of China was studied using a ToF-CIMS instrument with a Filter Inlet for Gas and AEROsols (FIGAERO) unit. NACs were classified based on their similarity to NP, NC, nitrobenzoic acid, and DNP. The measurements were

performed in Dezhou, China where open burning of crop residues is a major source of atmospheric pollutants (Wang et al., 2018). A previous study conducted in this area recorded some of the highest levels of emissions from open burning of crop residues ever observed in Shandong province (Gao et al., 2017), so it was expected that emissions due to BB events would be captured during the study period.


## 2. Experimental Design

### 2.1 Site Description

Measurements were conducted in Dezhou, Shandong Province, China (37.4341° N, 116.3575° E), as part of the "Photochemical Smog in China" project, which aims to evaluate haze formation in China and its implications for air quality policies (Hallquist et al., 2016). Instrumental measurements were performed at the Meteorological Weather Bureau of Pingyuan from November 2017 to January 2018. This season is of specific interest because previous wintertime measurements in this area have indicated that NACs can contribute as much as 50 ng m$^{-3}$ to the mass concentration of PM$_{2.5}$. The temperature and relative humidity during the measurement period ranged from -11.7 to 20.9 °C and from 12 to 99%, with campaign averages of 2.2 °C and 50%, respectively. The wind speed averaged 2.4 m s$^{-1}$; a time series of the metrological conditions during the experimental campaign is shown in Figure 1. The daytime mass concentration of particulate matter (PM$_{1.0}$) measured with an aerosol mass spectrometer typically exceeded the allowable limit for PM$_{2.5}$ (25 μg m$^{-3}$), and there were more than 10 pollution episodes with high aerosol loadings (> 100 μg m$^{-3}$). Organic matter comprised 60% of the PM$_{1.0}$ on average, and it contributed as much as 80% in several field measurements. Inorganic nitrate (NO$_3^-$) accounted for 20% of the measured PM$_{1.0}$ levels on average. The diurnal profiles of total particulate matter and organic matter were similar, with two distinct peaks at 8:00 AM and 7:00 PM (L.T.) (Figure 1).



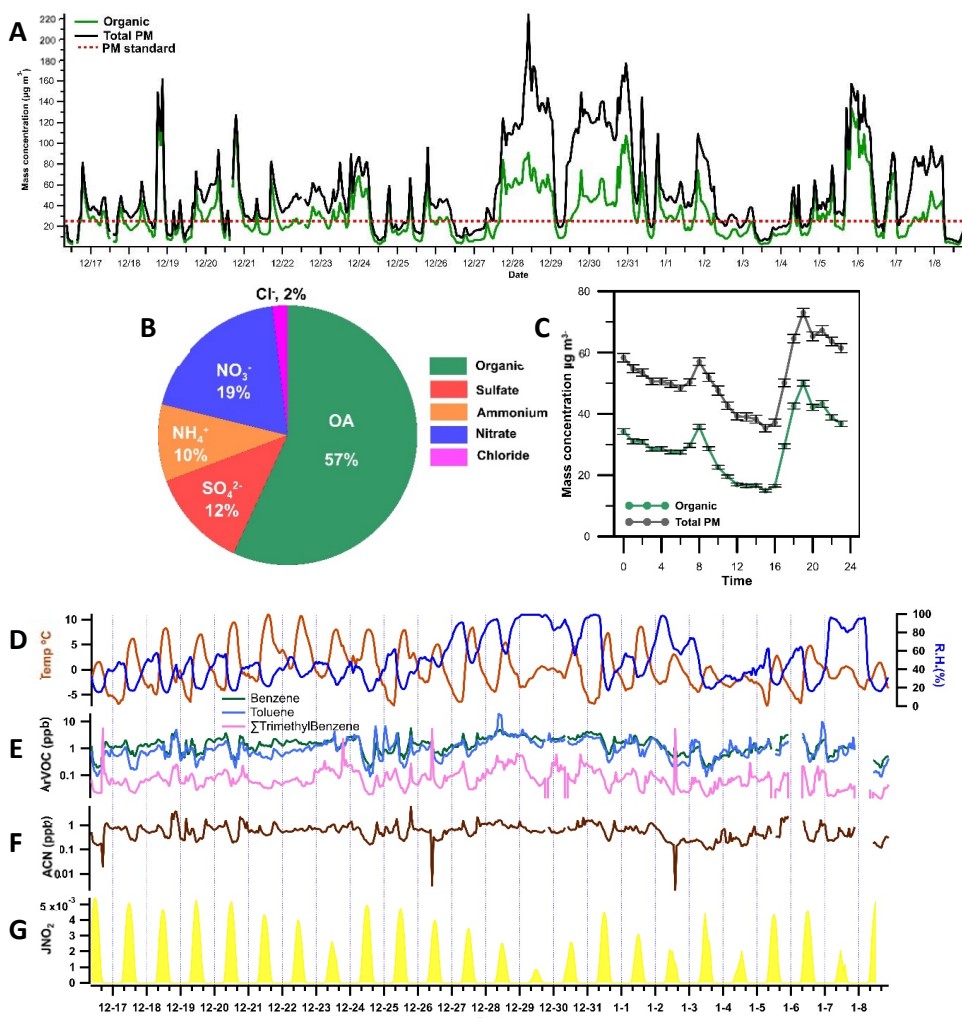

**Figure 1.** *(A) Time series profiles of organic matter (OM) and total particulate matter (PM) concentrations during the field campaign. The red reference line at 25 µg m⁻³ represents a typical PM₂.₅ mass limit. (B) Contribution of different aerosol components to particulate mass concentration as measured with an aerosol mass spectrometer (AMS). (C) Diurnal variation in levels of organic and total particulate matter during the wintertime measurement period in Dezhou. Error bars indicate standard errors. Time series of (D) temperature and relative humidity (RH), and (E) aromatic VOCs such as benzene, toluene, and trimethylbenzene. Also included in the figure are (F) the mixing ratio of acetonitrile (ACN) and (G) the rate of NO₂ photolysis, expressed as j(NO2).*

## 2.2 FIGAERO-CIMS measurement

A Filter Inlet for Gas and Aerosol (FIGAERO) time-of-flight mass spectrometer (ToF-CIMS)

was utilized to characterize the NAC content of the gas and particle phases. This instrument is described

5    in detail in previous publications (Lopez-Hilfiker et al., 2014;Le Breton et al., 2019;Le Breton et al.,



2018). Teflon tubing and copper tubing were used as sample lines for the gas and particle phases, respectively. The ToF-CIMS was operated in negative ionization mode with iodide (I-) as the reagent ion. High purity $N_2$ air (99.9% purity) was flown over a glass vial containing methyl iodide ($CH_3I$) and into a ToFwerk type P X-ray ion source (operated at 9.5 kV and 150 μA) to create ions to charge

compounds (MH) entering the ion-molecule region (IMR). The resulting product ions were identified either as molecular adducts with iodide ($MHI^-$) or deprotonated ions ($M^-$) (equation 1):

$$(H_2O)I^- + MH \rightarrow n(H_2O) + MHI^- /M^- \qquad \text{(eq. 1)}$$

The ToF-CIMS was optimized to have an average spectral mass resolution (m/Δm) of 3000. ToF spectra

showing ions with mass-to-charge (m/z) ratios between 7-620 Da were acquired with a time resolution of 1 s and averaged over 1 minute for data analysis The ToF- spectra were mass calibrated using four frequently occurring ion peaks: Iodide monomer ($I^-$, 126.904 m/z), dimer ($I_2^-$ 253.809 m/z), and trimer ($I_3^-$, 380.713 m/z), and the $NO_3^-$ ion (61.988 m/z).

Aerosol particles were collected for 30 minutes with a $PM_{1.0}$ cyclone and deposited on a

Zefluor® PTFE membrane filter. Analytes were then desorbed by passing heated $N_2$ gas over the filter, with a temperature cycle from room temperature to 200 °C over 20 minutes, followed by a 10-minute soak at 200 °C to ensure desorption of the compounds from the filter (a typical desorption profile is shown in the supplemental information). The NACs were quantified by doping the PTFE filter of the FIGAERO with known amounts of authentic standards. The standards were analyzed using the same

thermal desorption procedure as for the aerosol particles. NACs with no available standard were quantified by applying sensitivities for compounds with similar chemical structures.

**2.3. Other collocated instruments**

A high-resolution time-of-flight aerosol mass spectrometer (HR-ToF-AMS) was used to

measure the composition and size distribution of particles with diameters below 1.0 μm ($PM_{1.0}$). These measurements provided mass concentrations of particle-bound non-refractory species such as organics, sulfates, nitrates, ammonium, and chloride with a 4-minute average time resolution (DeCarlo et al., 2006). Volatile organic compounds (VOCs), including some precursors of NACs, were monitored using

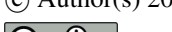



a combination of two online gas chromatographs (GC) with a mass spectrometer (MS) and flame
ionization (FID) detectors, resulting in the detection of over 100 VOCs. Photolysis rates of ozone
($j$(O1D)) and $NO_2$ ($J_{NO2}$) were measured using a commercial spectroradiometer, which was calibrated
using a high-power halogen lamp after the field campaign.

**2.4 Box model simulation of nitrocatechol using the AtChem tool**

A series of box model simulations was conducted to clarify the mechanism of NC formation during the
second period of the experimental campaign (see below), when secondary chemistry dominated the
formation of NACs. Simulations were performed using AtChem (Sommariva et al., 2020), an online
zero-dimensional box model, together with chemical reactions extracted from the Master Chemical

Mechanism (MCMv3.3.1) via the website http://mcm.leeds.ac.uk/MCM (Jenkin et al., 2003;Saunders
et al., 2003). AtChem was previously utilized to simulate the formation of formic acid and nitrophenol
at a site dominated by oil and gas production (Yuan et al., 2016). In our simulations, NC was used as a
representative NAC to clarify the potential contribution of compounds emitted during biomass burning
(i.e. catechol) to secondary NAC formation. The strong dependence of NC production on the overall

rate of secondary formation and its significant mixing ratio during the second period (74 ng m$^{-3}$) made
this compound a suitable representative NAC for this purpose. The MCM (v.3.3.1) assumes that NC is
the sole product of catechol, which greatly reduces the complexity of the model's calculations.
Measured concentrations of inorganic gases such as CO, $O_3$, $NO_x$, and selected volatile organic
compounds were used to constrain the simulations (Table S3). Over 90 VOCs were measured during

the field campaign in Dezhou. However, to minimize the computational cost of the modelling process,
only the 10 VOCs with the highest mixing ratios were included in the model. Additionally, compounds
with multiple isomers (e.g. xylene and trimethylbenzene) were treated as single species to further reduce
computational cost. The VOCs with the highest contribution to OH reactivity in four Chinese cities
(Tan et al., 2019) were also included in the model to properly simulate the major oxidation reactions in

the atmosphere. The VOC concentration assumed in the simulations amounted to 75% of the total VOC
concentration measured in Dezhou. Overall, a total of 1195 species (intermediates, products, etc.) and
3705 reactions (oxidation, photolysis, etc.) were included in the mechanism. The contribution of traffic
sources was also analyzed; results for these sources are presented in the supplemental information.



## 3. Results and Discussion

### 3.1 Identification and occurrence of NACs in Dezhou

Figure 2 compares mass spectra collected during a strong BB episode (red lines) to those for a typical clean day (black lines) in Dezhou. The intensities of several molecular ion peaks increased

significantly during the BB episode, particularly those at 138.019 and 154.014 m/z, which correspond to the deprotonated masses of NP ($C_6H_4NO_3^-$) and NC ($C_6H_4NO_4^-$), respectively. These ions exhibited six to eight-fold increases in signal intensity during the BB episode, clearly indicating substantial increases in the concentrations of the corresponding compounds. By analyzing the difference between polluted and clear episodes, 16 ions related to nitro-aromatic compounds were identified (see Figure 2),

including nitrobenzoic acid, methoxy/methyl NP, and DNP. The exact positioning/assignment of the functional groups of the nitro-aromatic compounds could not be determined because ToF-CIMS cannot differentiate between isomers, i.e. compounds with the same molecular formulas. High resolution fitting results for individual peaks are presented in the supplemental information.

The campaign average mixing ratios of NACs measured for the gas and particle-phase were

1720 and 299 ng m$^{-3}$, respectively. The measured fractions of the 16 NACs in the particle phase ($F_p$) ranged between 9 and 28%, with a mean of 16%; these results are consistent with those obtained in an earlier study that applied the same measurement technique during springtime at a suburban site in Changping near Beijing, China (Le Breton et al., 2018). The overall contribution of particle-phase NACs to the total concentrations of organic matter (mean:1.9%; range: 0.0025-21 %) and total PM

(mean:1.1%; range: 0.0013-11 %) varied substantially over the campaign, which may indicate that multiple NAC sources and formation pathways influence the concentration of NACs. Since the analytical technique used in this work cannot account for all NACs, the mass loadings reported here should be treated as lower limits. Nevertheless, the mean concentrations measured in Dezhou were higher than the mixing ratios of total NACs reported in other studies (Teich et al., 2017;Wang et al.,

2018;Wang et al., 2019;Kahnt et al., 2013), possibly because this work examined a greater number of NACs (16 compounds). Furthermore, concentrations of NACs measured in winter are normally significantly higher than those measured in the summer because the boundary layer is usually more shallow and BB (wood combustion) occurs more frequently in winter.



The diurnal profile of total gas-phase NACs exhibits two distinct peaks: a broad peak around midday, between 13:00-16:00, and another in the evening (around 20:00 local time), as shown in Figure 3. The mean daytime concentration of gas-phase NACs in Dezhou (2200 ng m$^{-3}$) was almost twice the night-time mixing ratio (1400 ng m$^{-3}$), presumably because either the rate of NAC production was higher during the daytime or the rate of loss was lower. These results stand in contrast to those of an earlier study, in which only night-time peaks were observed due to the daytime photolysis of NACs (Yuan et al., 2016). Mean day- and night-time concentrations of particle phase NACs were similar (304 and 300 ng m$^{-3}$, respectively) but with clear variability, as shown in the corresponding diurnal profiles: there were two diurnal particle-phase peaks (08:00 and 20:00 L.T.), but both were less pronounced than their gas phase counterparts. These maxima coincided with the peaks in the diurnal profiles of organic and particulate matter, suggesting that levels of particle-phase NACs were linked to the general occurrence of ambient aerosols. This may be due to enhanced partitioning towards the particle phase caused by increases in the organic aerosol mass.

Figure 3 also shows the mass contribution of lumped NAC categories in the gas and particle phases. NACs were assigned to lumped categories based on their structural similarity to the most common NACs reported in previous studies. Both the gas and particle phases exhibited similar percentage contributions for each category. NP and its analogs accounted for almost half of the total NAC concentration in both phases, which was assumed to be due to the strong influence of primary emission from BB events. NC and methylnitrocatechol, both of which are commonly used as biomass tracers (Iinuma et al., 2010;Finewax et al., 2018), individually accounted for as much as 9% of the total NAC concentration. Interestingly, the diurnal profiles of NP did not follow the general trend of the other measured NACs. This suggests that its formation pathway differs from that of other NACs such as NC; it may be that the contribution of secondary formation is greater than that of direct primary emission from BB for NP.



**Figure 2. (Top)** *Integrated mass spectra during a biomass burning episode (red) and a typical "clean" day.* **(Middle)** *Expansions of the peaks at m/z 138 and 154, which correspond to deprotonated nitrophenol (NP) and nitrocatechol (NC). The increase in the strength of the NP and NC signals during the biomass burning episode is readily apparent.* **(Bottom)** *Molecular structures of the 16 NACs identified in this work, grouped according to structural similarity.*



### 3.2 Sources of nitro-aromatic compounds

Previous studies indicated that NACs mainly originate from BB, traffic or secondary formation in the gas or condensed phases, with minor contributions from coal combustion (Hanson et al., 1983;Wang et al., 2018;Yuan et al., 2016;Xie et al., 2019;Chow et al., 2016). While previous studies found that traffic has important effects on NAC levels (Cecinato et al., 2005;Tremp et al., 1993), its influence appeared to be limited in this case: there was a weak to negative association between typical automobile exhaust VOCs (benzene, toluene, and trimethylbenzene) (Zhang et al., 2018;Geng et al., 2008;Batterman et al., 2002) and the studied NACs. Furthermore, concentrations of the measured NACs did not peak during or shortly after periods of high traffic intensity, in contrast to results obtained at three sites in Europe during cold and warm seasons (Delhomme et al., 2010).

NACs may also form in aqueous phases, particularly when the atmosphere has a high liquid water content (Harrison et al., 2005b). The contribution of aqueous phase oxidation to NAC formation was found to be limited based on the negative relationship ($r$ = -0.1 to -0.6) between the relative humidity (R.H.) and the mixing ratios of NACs in the gas and particle phases. An earlier study found that the relative contribution of aqueous-phase oxidation to NAC formation in Beijing increased as the ambient R.H. increased (Wang et al., 2019). The conditions in Dezhou during the measurement campaign were relatively dry (the campaign average R.H. was 50%), with few days exceeding 70% R.H. as shown in Figure 1.

Given the minor contributions of traffic and aqueous phase oxidation, most of the subsequent analysis focused on primary BB emissions and secondary formation in the gas and particle phases.


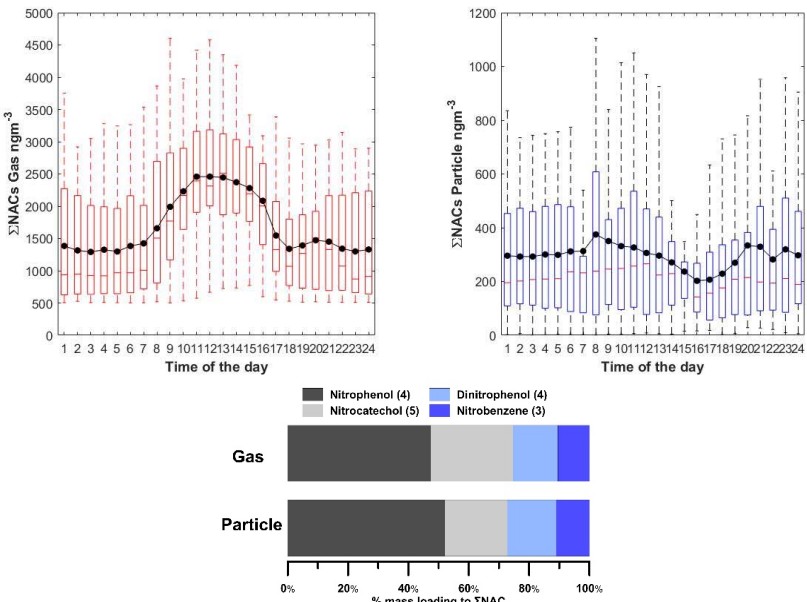

*Figure 3. (Top) Diurnal profile of the sum of NACs in the gas (left) and particle (right) phases. The solid black line indicates the hourly average. Note that outliers are not shown in the figure. (Bottom) Molecular distribution of NACs in the gas and particle phases, classified according to similarity to major NACs. Values in parenthesis indicate the number of compounds per category.*

## 3.3 Primary emission from biomass burning

The measurement period included many intense BB episodes that increased the concentrations of some NACs, particularly in the particle phase. To verify the association between these elevated concentrations and the observed BB episodes, the atmospheric behaviors of levoglucosan, nitrous acid (HONO), and the ratio of $NO_x$ to $NO_y$ ($NO_x/NO_y$) were used as tracers of BB. Levoglucosan is a commonly used molecular tracer of BB that is superior to other markers such as $K^+$ and black carbon (BC) because it is much less prone to interference from non-BB sources (Zhang et al., 2012;Simoneit et al., 1999). Concentrations of HONO increase during BB episodes because the rate of conversion of $NO_2$ into HONO is elevated in BB plumes due to the presence of aerosols with high surface areas (Nie et al., 2015). The ratio of $NO_x$ and $NO_y$ reflects the freshness of the emissions from combustion; levels of $NO_x$ in plumes originating from local BB are high, typically resulting in a ratio close to 1. The atmospheric transformation of $NO_x$ leads to the formation of $NO_z$ components such as $HNO_3$ and organonitrates, causing the ratio of $NO_x$ to $NO_y$ to deviate from unity. In Figure 4, the three-week measurement campaign is separated into four periods corresponding to four distinct NAC formation



regimes. Regimes 1 and 3 are associated with strong BB episodes based on the profiles of the three previously mentioned tracers. Levels of NACs in the condensed phase mirrored those of levoglucosan, which exhibited two strong peaks at 8:00 and 20:00 L.T. Moreover, during apparent BB events, the timing of the peaks in NAC concentrations agreed well with that of the peaks in the diurnal OM profile.

This demonstrates the apparently strong contribution of NACs to sub-micron aerosols in the studied city in rural China

Figure 4 also shows the gas and particle concentration time series of three representative NACs (NP, NC, and DNP). NP and NC are frequently reported to be the dominant NACs in field- and laboratory-based BB studies (Xie et al., 2019;Wang et al., 2017;Wang et al., 2018).  Figure 4 clearly

shows that the three NACs behave in quite different ways under the regimes linked to strong BB episodes. In these regimes, gas phase NP concentrations correlated strongly (r = 0.8 – 0.9) with those in the particle phase. This may be because there was a common dominant source of NP in the gas and particle phases, or because of fast partitioning of NP between these phases during BB events. There was also good agreement between the gas- and particle-phase time series for the methoxy/methyl ($C_7H_7NO_3$)

and ethoxy/ethyl ($C_8H_9NO_3$) derivatives of NP during BB regimes. NP and its analogs thus appear to be good direct tracers of primary emissions from BB events in Dezhou.

Conversely, the correlations between the gas and particle phase concentrations of NC and DNP were very weak (r = 0.2 – 0.3), indicating that BB events had different effects on the formation and partitioning of these NACs (see supplemental information for a complete correlation analysis). During

typical clean days, particularly from Dec. 20 to 30 (regime 2), the average correlation coefficient between the gas and particle phases for all NACs fell to less than 0.5, possibly because the contribution of photochemical processes to NAC formation was high relative to that of primary BB sources.

Secondary NAC formation may also occur during periods of extensive BB, resulting in mixed contributions to the observed NAC concentration. Differences in the mixing ratios and mass

concentrations of NACs between BB regimes and relatively clean regimes can shed light on the relative contributions of primary emissions from BB and secondary production during each regime type. This can be demonstrated by considering the average NAC concentrations under regimes 1 (a strong BB regime) and 2 (a non-BB regime); pronounced differences (>50%) in NAC concentration between these



regimes can be considered indicative of the influence of BB on NAC production in Dezhou. The average recorded signal intensities for levoglucosan in the gas and particle phases under regime 2 were 52 % and 72% lower, respectively, than those for regime 1. Moreover, some compounds, particularly NP and its methoxy/methyl and ethoxy/ethyl derivatives, exhibited significantly reduced mixing ratios in both

5    gas and particle phases under regime 2. This supports the position that these compounds may be useful direct tracers of BB in Dezhou. Some NACs exhibited lesser declines (<30%) in the gas phase mixing ratio, suggesting that BB may not be their dominant source; for example, they may be primarily formed via secondary production. This behavior was observed for dinitrated aromatic compounds such as $C_6H_4N_2O_5$ and $C_7H_6N_2O_5$, which can be formed via nitration of mononitrates ($C_6H_5NO_3$ and $C_7H_7NO_3$)

10    (Yuan et al., 2016;Vione et al., 2005). Such nitration processes are mainly driven by secondary photochemical or multiphase reactions, explaining the comparatively small difference in the mixing ratios of these compounds between regimes 1 and 2.



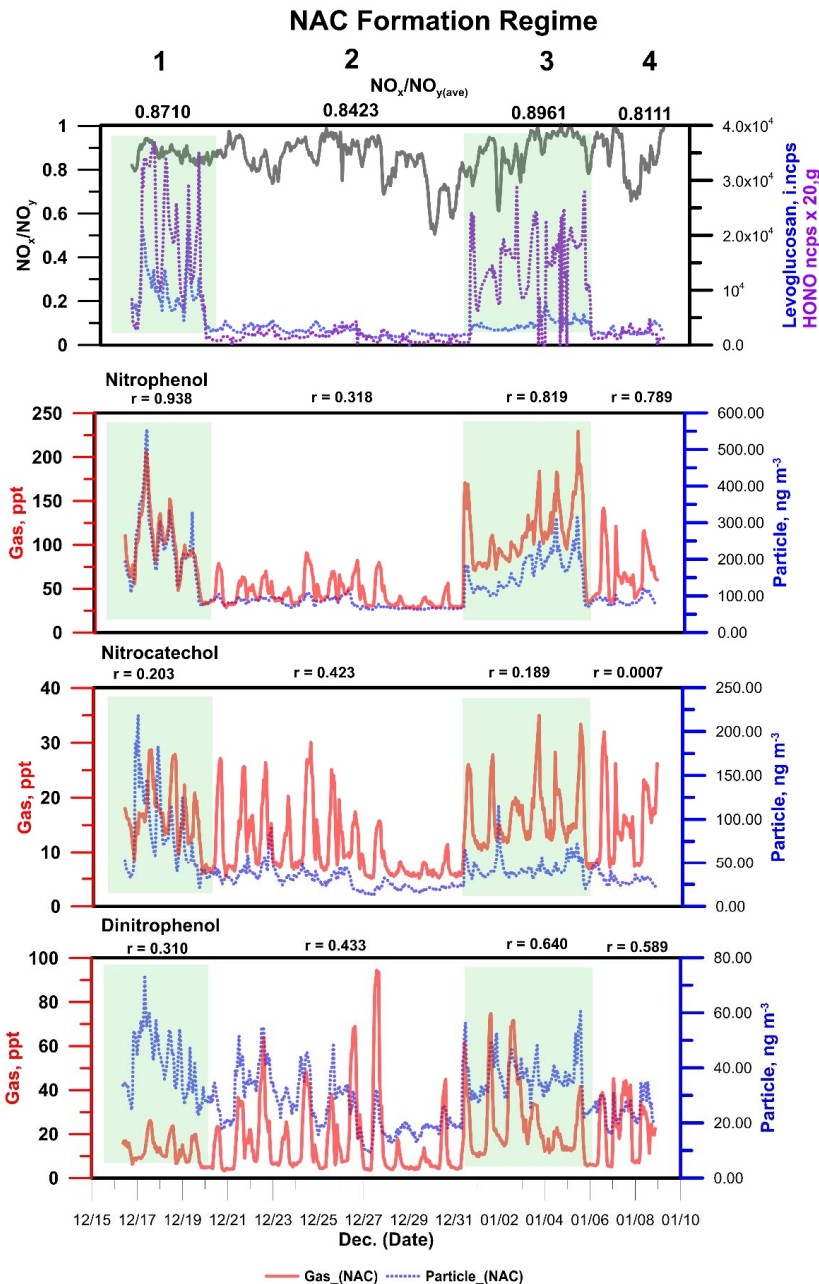

***Figure 4 (Top).*** *Time series of particle phase concentrations of levoglucosan, gas phase concentrations of HONO, and ratios of $NO_x$ to $NO_y$ as markers of BB episodes during field measurements.* ***(Bottom three panels)*** *Mixing ratios of nitrophenol, nitrocatechol and dinitrophenol under regimes corresponding to BB episodes and non-BB episodes. The coefficient of correlation (r) reflects the agreeement between the gas and particle phase concentrations.*



### 3.4     Secondary formation of gas-phase NACs

Under regime 2, concentrations of BB markers fell dramatically, indicating that the influence of primary biomass emissions was limited (as shown in Figure 4). Here, the diurnal profiles of gas-phase NACs (as shown in Figure 5) exhibit increases in concentration at 14:00-15:00 L.T. and a minor night-time peak at 20:00 L.T. Similarly, in contrast to the events during regime 1, the peak in particle phase NAC concentrations occurred also in the afternoon 14:00-15:00 L.T. These peaks in the daily mixing ratios NACs coincided with the daily peak ozone concentration. Secondary photochemical formation was therefore probably the dominant NAC formation process under regimes 2 and 4. This conclusion is supported by the fact that the coefficient of determination ($r^2$) between ozone and nitrophenol (see Figure 5) under regime 2 ($r^2=0.7$) is substantially higher than that for the full dataset including BB regimes 1 and 3 ($r^2=0.1$). The most pronounced reductions in $r^2$ were observed for compounds expected to originate mainly from primary sources (e.g. NP); for compounds expected to be formed mainly via secondary production (e.g. NC and DNP), the $r^2$ with ozone remained relatively high throughout the campaign (see Figure 5).

Secondary production of NACs can be linked to the presence of specific precursor compounds (Harrison et al., 2005a). Figure 6 shows the correlations between levels of NP, NC, and DNP and those of their proposed precursors - phenol ($C_6H_6O$), catechol/dihydroxybenzene ($C_6H_6O_2$), and NP. Phenol and catechol are primarily formed by the pyrolysis of lignins and can be precursors for secondary formation of NACs, particularly during BB events (Yee et al., 2013;Finewax et al., 2018;Gaston et al., 2016). Levels of NACs correlated strongly ($r^2=0.7-0.8$) with those of their primary precursors (i.e. phenol and catechol). This indicates that nitration of these precursor phenolic compounds in the presence of OH or $NO_3$ radicals was an important route of NP and NC formation. The figure showing the correlation between precursors and final products also shows the observed ozone mixing ratio, which is a measure of secondary photochemical activity. This further underscore the significance of photochemical oxidation in the formation of NC and NP from catechol and phenol. A similar relationship was observed for the secondary formation of DNP via further oxidation of nitrophenol. DNP is formed by the reaction of nitrophenol with OH or $NO_3$ radicals to form nitrophenoxy radicals ($NO_2C_6H_5O^•$) whose subsequent nitration yields DNP (Yuan et al., 2016).


As shown in Figure 4, the ratio of $NO_x$ to $NO_y$, which is an indicator of plume freshness, was lower under regime 2 than regime 1, suggesting that an older plume was sampled in the former case. This aged plume may have contained residual traces of regional photochemical smog containing phenol, catechol, and their derivatives that were formed as primary emissions during BB events outside the studied region.

The yield of NACs produced by secondary formation is known to depend on the $NO_2$ concentration (Wang et al., 2019;Yuan et al., 2016;Wang et al., 2018). For instance, NP is formed by the nitration of phenoxy radicals ($C_6H_5O^\bullet$), which are themselves formed by the $OH/NO_3$-mediated oxidation of phenol (Berndt and Böge, 2003). Mechanistically, the formation of NACs such as NP should be heavily dependent on the atmospheric concentration of $NO_2$. However, NACs such as NP and nitrosalicylic acid were formed consistently in a mountainous region of China even when the $NO_2$ concentration was below 5 ppb (Wang et al., 2018). The campaign $NO_2$ average for this work in Dezhou was 23 ppb, with day- and night-time means of 17 and 26 ppb, respectively. These mixing ratios may have been high enough to sustain the nitration of aromatic VOCs. However, a negative correlation was observed between $NO_2$ and NACs in the gas ($r_{ave.}$ = -0.598) and particle phases ($r_{ave.}$ = -0.116) under regime 2, when secondary formation was the dominant source of NACs. In aged air masses such as those sampled during regime 2, $NO_x$ will be transformed into nitrated compounds (and $HNO_3$), which may explain this negative correlation.



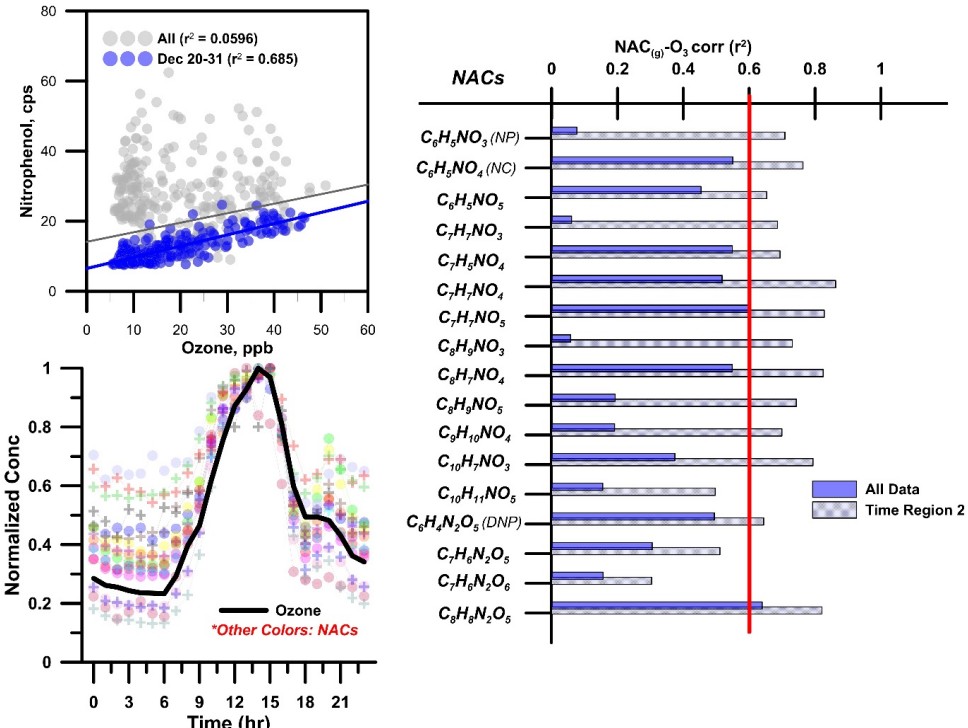

***Figure 5 (Top left)*** *Correlation between levels of nitrophenol and ozone (O₃)* ***(Bottom left)*** *Normalized diurnal profiles of gas phase NACs and ozone under the second NAC formation regime observed during the field campaign* ***(Right)*** *Coefficients of determination ($r^2$) between gas phase NACs and O₃ for the whole data set and for regime 2 only.*



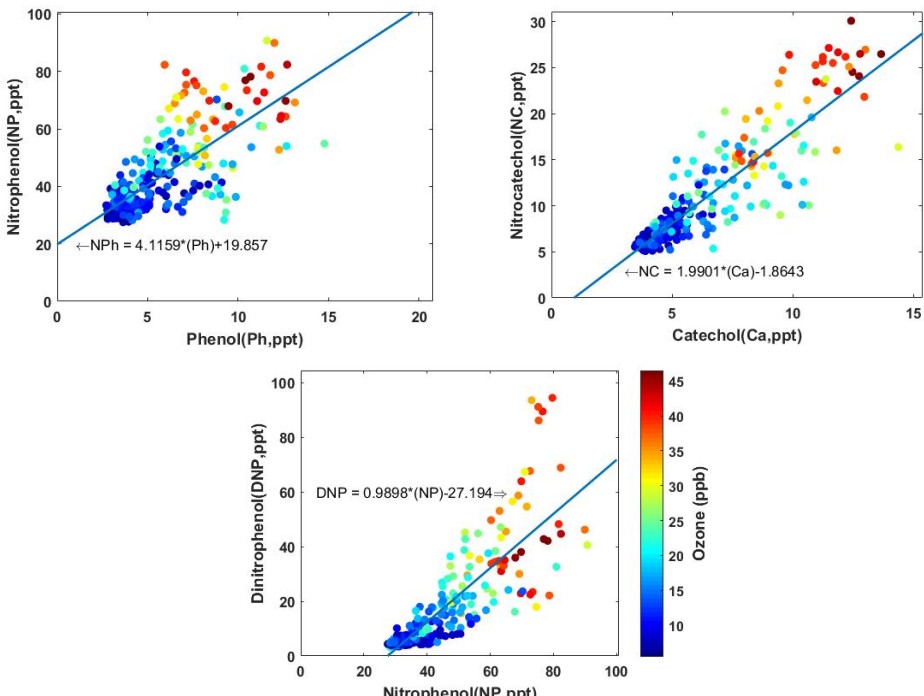

***Figure 6.*** *3D scatter plots showing the variation of the concentrations of NP, NC and DNP with the concentrations of their precursors together with the corresponding ozone mixing ratios.*



### 3.5    Analysis of NAC production and loss pathways

*Figure 7. Schematic depiction of the atmospheric formation and loss of nitrocatechol in the gas phase based on the reaction pathways included in the Master Chemical Mechanism (MCMv3.3.1). Species names used in the MCM are given in parentheses.*

To further investigate the secondary production of NAC during the experimental campaign, box-simulations were performed to model NC formation and loss using the AtChem tool and atmospheric oxidation chemistry models from MCMv3.3.1. Figure 7 shows the reaction pathway for the formation of NC by catechol oxidation initiated by OH or $NO_3$ radicals. Unlike in the case of NP, only one precursor – catechol – can generate the intermediates (i.e. CATEC1O, CATEC1O2, and CATEC1OOH) in NC formation. Sinks of NC are its further oxidation by $NO_3$ or OH radicals, which lead to stable ring-opening products such as 2-oxoacetic acid. Photolysis and deposition/dilution of NC were also accounted for in the simulation because of their reported importance in the gas-phase atmospheric loss of formic acid and NP (Yuan et al., 2015;Yuan et al., 2016). The photolysis frequency of NC used in the simulations was based on the reported value for NP (1.4% of the photolysis frequency of $NO_2$). A sensitivity analysis (see supplement) of the box model against variation of the effective physical loss rates (due to dispersion and deposition) indicated that a high loss rate (1 hour) provided the best estimate of the observed NC mixing ratios. Physical loss terms with equivalent lifetimes above 1 hour (e.g. 3 hours) overestimated the measured NC concentrations by at least 50%. As a result, the tail of the modeled daytime peak extended well into the night when using low physical loss rates. The loss rate used in this work was higher than those used in previous box model analyses of formic acid and NP (Yuan et al., 2015;Yuan et al., 2016) but is reasonable given the low vapor pressure of NC (2.1 × $10^{-4}$ Pa) (Finewax et al., 2018), which favors partitioning into the condensed phase





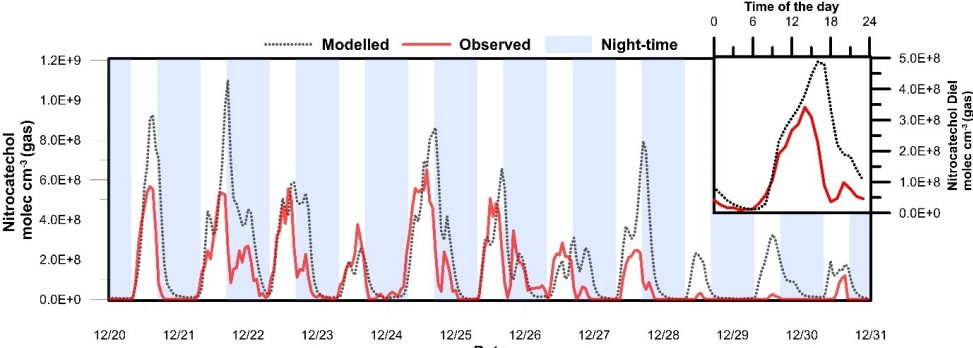

**Figure 8.** *Modeled and observed nitrocatechol concentration time series.* **(Inset)** *Diurnal profile of the observed and modeled nitrocatechol concentrations.*

Figure 8 shows time series of the observed and modeled mixing ratios of nitrocatechol under regime 2. The simulated mixing ratio profile agrees reasonably well with the experimental data, as indicated by the mean ratio of the modeled concentration to the observed concentration (Model/Obs$_{ave}$ =1.25) and the coefficient of determination ($r^2 = 0.51$) between the two data sets. These results clearly show the explicit dependence of the secondary formation of NACs such as nitrocatechol on the oxidation of thermal degradation and pyrolysis products of lignins (e.g. catechol) in aged plumes. Additionally, the presence of the simulated daytime peak confirms that the rate of daytime production of NC (source) exceeded its rate of photolysis (sink) during the second period of the field campaign. If the daytime loss rate of NC due to photolysis is disregarded, the mixing ratio will only increase by 10%, clearly showing the weak contribution of photolysis to the overall loss of NC. The primary pathways of NC loss were thus oxidation by OH radicals and night-time oxidation by $NO_3$ radicals.

Figure 8 also shows the diurnal profiles of the modeled and observed NC concentrations. The modeled profile features prominent peaks at 10:00 (shoulder), 16:00, and 20:00 local time, but the experimentally observed afternoon peak in NC levels occurred around 14:00 L.T. This discrepancy can be explained by the change of wind direction from northwest to northeast observed after 14:00 L.T. It should be noted that the parametrization of AtCHem does not account for meteorological effects, and that it only partially accounts for dispersion via the effective physical loss rate parameter. The three daily maxima were attributed to the contributions of different sources of the intermediate



hydroxyphenoxy radicals (CATEC1O) throughout the day. As shown in Figure 7, the nitration of

CATEC1O radicals is the only source of NC, so the production of hydroxyphenoxy radicals will dictate

the overall rate of NC formation under excess $NO_x$ conditions. Figure 9 shows the relative contributions

of the three major CATEC1O formation pathways. Note that CATEC1O can also be produced through

photolysis of hydroperoxylphenol (CATEC1OOH) and the reaction of hydroxyphenylperoxy

(CATEC1O2) with $NO_3$ and $RO_2$ radicals, as shown in Figure 7. However, these pathways account for

less than 0.05% of the total CATEC1O production and were therefore disregarded. The daytime

shoulder peak of NC at 10:00 was due to OH-radical oxidation of catechol and accordingly coincides

with the diurnal peak in the OH concentration ($\sim$4.5 x $10^6$ molec. cm$^{-3}$). The major formation pathways

of OH radicals in Dezhou during wintertime were HONO photolysis and the reaction of $HO_2$ with NO,

causing measured OH production to peak in the afternoon rather than at midday as it is more common.

Ozone photolysis, which is typically a major source of OH radicals, made a negligible contribution

under the studied wintertime conditions. Additionally, the box model indicated that the high levels of

CATEC1O at 13:00 were predominantly due to the reaction of NO with hydroxyphenylperoxy radicals

($\sim$50%). Finally, the elevated levels of NC at 20:00 were primarily attributed to the very efficient ($\sim$90%

conversion) $NO_3$ nighttime chemistry after sunset (16:30).

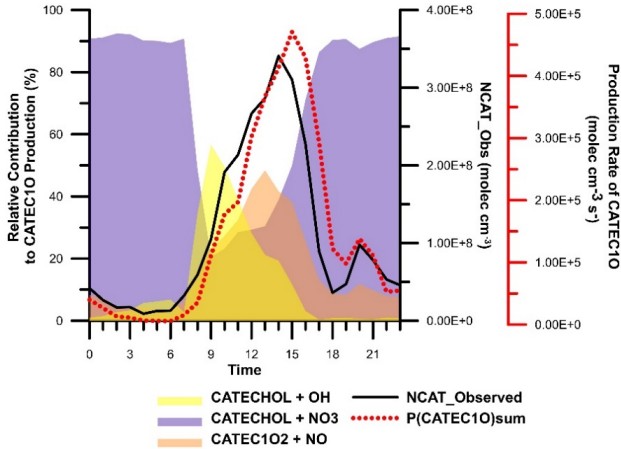

***Figure 9.*** *Diurnal variation in the relative contribution of the hydroxyphenoxy (CATEC1O) pathway to overall nitrocatechol formation. Also shown are the observed concentrations of nitrocatechol and the summed rates of*
*CATEC1O production.*

**3.6     Classification and quantification of sources of nitro-aromatic compounds**



The general sources of NACs have been explored in previous works (Wang et al., 2019;Wang et al., 2017). The 16 NACs found in Dezhou were further categorized based on their main formation routes: primary BB and secondary formation. As noted above, secondary processes were dominant under regime 2 whereas both routes contributed under regime 1. The NACs detected under regime 1 were further classified based on the correlation between their gas phase and condensed phase concentrations. A strong correlation between the gas and particle phase ($r_{g/p}$) was taken to indicate either that primary BB was the dominant source of the NAC in question or that it underwent rapid partitioning between phases. Five NACs exhibited $r_{g/p}$ values above 0.75 and thus behaved like NP. The gas and particle phase concentrations fell by at least 55% and 85%, when the major source of NACs shifted from primary BB under regime 1 to secondary formation under regime 2. However, these five compounds were also formed under regime 2, suggesting that secondary formation does contribute to their presence. To assess the impact of each formation pathway under regime 1, two approaches were used. In the first approach, it was assumed that the degree of secondary formation was similar under regimes 1 and 2, which is reasonable based on the average ozone levels under each regime ($O_{3ave} = 20$ ppb). Both regimes had similar total gas and particle-phase concentrations of DNP, a product only formed by secondary oxidation, further supporting the validity of this assumption. This first approach was referred to as the DNP method based on the similar DNP profiles observed under regimes 1 and 2. By subtracting the concentrations of the five BB compounds under regime 2 from those under regime 1, it was determined that primary BB combustion processes accounted for 70% of the observed concentrations of these compounds. The second approach used to estimate the contribution of primary BB to the measured NAC concentrations involved using levoglucosan as a primary source tracer. This approach is analogous to the widely used EC tracer approach, in which elemental carbon (EC) is used to distinguish the primary organic carbon (POC) fraction from secondary organic carbon (SOC) in total organic carbon (OC) measurements (Day et al., 2015;Cabada et al., 2004). The high time resolution levoglucosan (lev.) measurements were performed using the same instrument and conditions as the NAC measurements, so they provided good data coverage, making lev. a suitable tracer of primary BB. The relative contributions of primary emission (BB) and secondary (sec) formation for each NAC were estimated using the following expression:



$$[NAC]_{BB} = ([NAC]/[lev.])_{BB} \times [lev.] \qquad (eq.\ 2)$$

$$[NAC_{sec}] = [NAC_{Tot}] - [NAC_{BB}] \qquad (eq.\ 3)$$

Here, $([NAC]/[lev.])_{BB}$ is the ratio of the concentration of the NAC to that of lev. during strong primary combustion emission, and $NAC_{BB}$ and $NAC_{sec}$ are the fractions of NACs generated through biomass

burning and secondary production, respectively. $NAC_{Tot}$ and lev. are the measured concentrations of NACs and levoglucosan in ambient measurements, respectively. Using this approach, primary BB combustion processes were found to account for 60% of the total production of BB-related compounds under regime 1, in good agreement with the estimate obtained using the DNP method.

The secondary compounds were categorized as such based on the weak correlations between their gas

and condensed phase concentrations ($r_{g/p} < 0.5$) under regime 1 and their association with ozone ($\Delta r^2(NAC\text{-}O_3)$), which is indicative of formation via secondary chemistry. For NACs mainly formed via secondary oxidation (e.g. nitrocatechol), the coefficient of determination ($r^2$) between ozone and NACs under regime 2 was similar to that for the full data set. This suggests that the formation pathways of these compounds did not change over the measurement period, regardless of the occurrence of

combustion episodes. Table 1 shows the classifications of 13 of the 16 NACs examined in this work, including their total concentrations under regimes 1 and 2. The classifications were further supported by the correlations between the concentrations of the 16 NACs (Figure 10), which clearly divide the NACs into different groups based on their behaviour in the gas phase under regime 1. The data in Table 1 was used to estimate the relative contributions of primary and secondary processes under the two

regimes (Figure 10), revealing that the relative abundance of NACs associated with secondary formation processes increased to 40% in both the gas and particle phases under regime 2.



**Table 1.** *Classification of the detected nitro-aromatic compounds based on their major formation pathways*

| NACs | $r_{g/p}$ § | $\Delta r^2$(NAC-O$_3$)$^\&$ <0.5 | Total conc. (g+p)$^\$$ | | Class | Primary contribution from BB in regime 1 (2) | |
|---|---|---|---|---|---|---|---|
| | | | Reg 1 | Reg 2 | | DNP. % | Lev. % |
| $C_9H_9NO_4$ | ✓ | | 190 | 51.7 | BB | 73 | 58 (46) |
| $C_8H_9NO_5$ | ✓ | | 118 | 35.5 | BB | 70 | 60 (48) |
| $C_8H_9NO_3$ | ✓ | | 381 | 62.2 | BB | 84 | 68 (40) |
| $C_7H_7NO_3$ | ✓ | | 558 | 121 | BB | 78 | 67 (53) |
| $C_6H4NO_3$ (NP) | ✓ | | 786 | 290 | BB | 63 | 64 (76) |
| $C_{10}H_7NO_3$ | X | ✓ | 40.7 | 23.7 | S | - | - |
| $C_8H_8N_2O_5$ | X | ✓ | 67.6 | 43.4 | S | - | - |
| $C_8H_7NO_4$ | X | ✓ | 49.4 | 32.9 | S | - | - |
| $C_7H_7NO_5$ | X | ✓ | 83.2 | 49.9 | S | - | - |
| $C_7H_6N_2O_5$ | X | ✓ | 41.7 | 35.2 | S | - | - |
| $C_6H_5NO_5$ | X | ✓ | 53.9 | 30.1 | S | - | - |
| $C_6H5NO_4$ (NC) | X | ✓ | 180 | 85.1 | S | - | - |
| $C_6H_4N_2O_5$ (DNP) | X | ✓ | 137 | 148 | S | - | - |
| $C_7H_7NO_4$ | | ✓ | 253 | 101 | UNC | - | - |
| $C_7H_6N_2O_6$ | | ✓ | 77 | 47.4 | UNC | - | - |
| $C_7H_5NO_4$ | | ✓ | 66.5 | 59.6 | UNC | - | - |

§$r_{g/p}$ =*Correlation coefficient between the gas and particle phases under regime 1 (✓- >0.75; X<0.5)*
&$\Delta r^2$*(NAC-O$_3$) = difference between the correlation of determination between NAC and ozone under regime 2 and that for the complete data set.*
$^\$$*g = gas; p=particle*
**DNP** = *Dinitrophenol method; Lev. = levoglucosan tracer method, values in parentheses are for regime 2*
*BB= Biomass Burning; S=Secondary formation; UNC= unclassified*

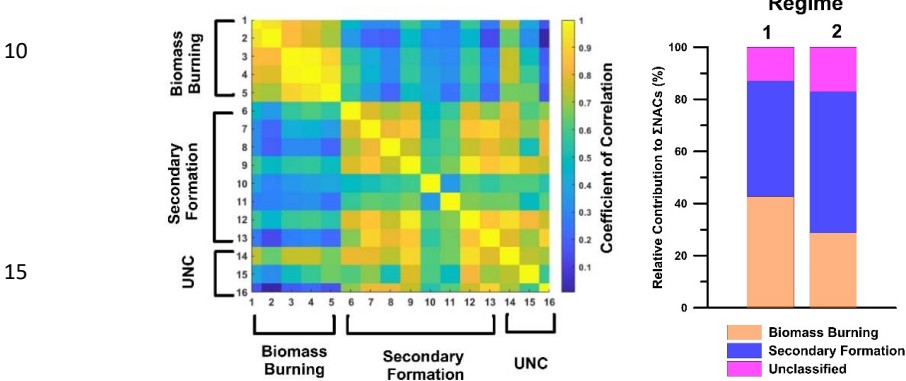

**Figure 10.** *(Left) Correlation matrix for gas-phase NACs under regime 1. (Right) Relative contributions of the identified NAC classes under regimes 1 and 2.*





**4. Conclusion and atmospheric implications**

High concentrations of 16 NACs were observed during an aerosol measurement campaign using a FIGAERO-ToF-CIMS at a rural site in China during wintertime where biomass burning is predominant. The mean overall NAC concentration during the measurement period was 2020 ng m$^{-3}$,

with nitrophenol and its analogs accounting for around half of this total. NACs accounted for 1.8% and 1.1% of the total mass of airborne organic matter and total PM, respectively. These results can be compared to measurements performed downwind of London (Mohr et al., 2013), where the mean ratio of the mass of NACs to that of OM (the ΣNACs/OM ratio) was only about 0.5%. The very high contribution of NACs in Dezhou can be attributed to the extensive use of biofuels such as wood and

straw for cooking and heating among the rural population of China, particularly during the cold season (Chen et al., 2017). Surveys of the studied area during the field campaign accordingly revealed heavy use of wood fuels when cooking, which coincided with strong peaks in the measured levels of nitro-aromatic compounds and organic matter.

The measured levels of NACs in the gas and particle phases revealed that on average, 16% of

the formed NACs were present in the particle phase ($F_p$). The gas phase levels of some NACs correlated strongly with their particle phase concentrations, suggesting rapid gas to particle partitioning of primary emissions. Conversely, some typical secondary products exhibited weaker correlations between the two phases, presumably due to the contribution of strong gas-phase sources.

The measurement campaign was divided into four different periods associated with different

NAC formation regimes; in some periods, primary BB was the dominant source of atmospheric NACs, while in other periods secondary formation processes played a greater role. The contributions of other sources such as traffic and aqueous phase processes were found to be negligible. The mixing ratios of NP and its derivatives increased markedly under regimes coinciding with primary BB events, indicating that these combustion processes contribute strongly to the formation of nitrophenolic compounds. The

concentrations of gas and particle-phase NACs decreased by 40 and 60% on average (max: 80 and 88%) upon shifting from a regime dominated by primary emission to one dominated by secondary formation,





clearly indicating that the contributions of fresh biomass burning events to NAC formation significantly outweigh those of secondary formation processes.

During periods with low concentrations of biomass burning indicators such as levoglucosan, the diurnal profiles of gas-phase NACs clearly mirrored those of the atmospheric ozone concentration.

This was attributed to dominant NAC formation via regional secondary chemistry (photochemical smog). Under these conditions, the concentrations of precursors such as catechol and phenol correlated strongly with those of the corresponding secondarily produced NACs (e.g. NC and NP). Secondary formation was scrutinized using box-model simulations of NC formation and loss to link the observed results to specific chemical mechanisms. The three peaks in the experimentally observed diurnal NC

concentration profiles were attributed to variation in the production of the hydroxyphenoxy radical intermediate over the course of the day. The daytime increase in NAC levels observed in this work was higher than in previous studies, possibly because of a high production rate relative to the rate of primary loss via OH-mediated oxidation, which in turn was considerably higher than the rate of loss via photolysis (which accounted for only 10% of the observed loss).

The individual NACs identified here were classified and quantified based on the trends in their gas- and condensed-phase mixing ratios. Five of the 16 NACs, mainly the NP analogs, were classified as typical primary products of BB, while 8 were classified as originating mainly from secondary production. However, it was clear that secondary formation processes also contributed appreciably to the observed concentrations of the five primary BB compounds. These contributions were estimated

using two approaches - the dinitrophenol (DNP) and levoglucosan tracer methods. Both methods indicated that primary combustion processes accounted for at least 60% of the total production of the five NACs classified as primary BB products during the regime dominated by primary BB. However, the contribution of secondary processes was clearly significant, which may explain the high observed levels of other secondary compounds. This suggests that photochemical smog plays a significant role

in NAC formation even during wintertime air pollution events in rural China that are dominated by primary emissions.

Continuous and concurrent measurement of NACs in the gas and particle-phases with high time resolution has enabled us to understand the formation of compounds that induce radiative forcing in the atmosphere by absorbing near-UV and visible light. The finding that primary emission and secondary formation were the dominant NAC-forming processes during the measurement period highlights the

dominant contribution of combustion of vegetation and indoor fuels to the warming effect of brown carbon in the atmosphere. Additionally, the NAC production and loss pathways identified in this work provide insight into the climate impact of brown carbons, which partially depends on the lifetime of their light-absorbing components (Hems and Abbatt, 2018). Some oxygenated VOCs emitted during BB (e.g. catechol) were shown to have extended atmospheric lifespans, allowing them to persist beyond

the combustion events in which they were formed and serve as precursors for secondary formation of NACs, further exacerbating the warming effects of BB episodes. In this work, NACs were found to comprise less than 2% of the sampled aerosol by mass, so one might expect their overall impact to be minimal. However, heavy pollution episodes ($PM_{mass}$>100 µg m$^{-3}$) such those caused by biomass burning during wintertime may increase the formation of NACs that can induce climate forcing and

pose health hazards. There is thus a need for further research on the mechanisms of oxidation of these anthropogenic aromatic compounds under different scenarios, potentially building on the classification and quantification of NACs presented in this work, to better understand their global budgets and roles in climate forcing.

*Data availability.* The data used in this publication are available to the community, and they
can be accessed by request to the corresponding authors.

*Author contributions.* MaH, MiH, and SG were the project leaders for this measurement campaign. CMS, MLB, RT, and HW operated the CIMS. CMS, LL, MP, and ET performed the calibration procedures and modelling/simulation experiments. WZ, LZ, and YY supported with the analysis of other inorganic and organic pollutants. CMS and MaH wrote the paper. All authors
commented on the paper and were involved in the scientific interpretation and discussion.

*Competing interests.* The authors declare no competing interests.



*Acknowledgements.* The work was done under the framework research program on "Photochemical smog in China" financed by the Swedish Research Council (2013-6917). In addition, the National Natural Science Foundation of China (21677002), the National Key Research and Development Programme of China (2016YFC0202003), Swedish Research Council (2018-04430). are

5    acknowledge for financial support.



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
