# Peer review of "Ambient Nitro-Aromatic Compounds - Biomass Burning versus Secondary Formation in rural China"

_Atmospheric Chemistry and Physics, 2020_

## Referee Comment (RC1) · Anonymous Referee #1 · 9 Nov 2020

General:

This paper provides measurements of nitro-aromatic compounds in rural China using FIGAERO-TOF-CIMS and provides insight into the formation mechanisms of different compounds. This paper is well-written and a nice addition on this particular topic. I recommend publication after consideration of my comments.

Major Comments:

1. I suggest that the authors perhaps bring up occurrence of high sulfate haze events in China during winter, which has received a lot of attention, as additional motivation for better understanding more of these surprising secondary processes that dominate

PM in China during winter. I think this will broaden the readership of their work.

2. I suggest that the authors provide more information regarding the calibrations in the SI. Calibrating this instrument for NACs is challenging and the atmospheric community would benefit from a detailed description of how the authors performed their calibrations.

Specific Comments:

Introduction:

1. Page 3, Line 28 has a few grammatical issues. Remove "causes" and also "to" in between "results" and "strong"

2. Page 4, Line 16: please also cite [Lee et al., 2014]

Methods:

1. Section 2.1: I'd be curious to see HYSPLITs and also know if any information is available regarding boundary layer heights—both of which I think will help interpret the data. As the authors correctly point out, the AtChem model does not factor in meteorology and adding this information will help augment the chemical interpretation of their work.

2. Page 5, Line 15: is this a WHO limit or a national limit for PM2.5? Please clarify.

3. Page 6, line 3, please rename to Filter Inlet for Gases and Aerosols (FIGAERO) coupled to a chemical ionization time-of-flight mass spectrometer (TOF-CIMS).

4. Page 7, were any reference gases added to monitor drift in signal? Were any gas-phase zeros performed?

5. Page 7, Line 21: some of these compounds are light sensitive, were they used fresh for calibration?

Results:

1. Figure 2: I suggest pointing out the levoglucosan peak in the spectra and other markers used to indicate BB events.

2. Page 10, Line 9: your diurnal peaks in particle phase NACs are similar to those from residual wood burning seen in Gaston et al 2016.

3. Page 17, Lines 20-22, I'd be curious to know if the secondary production of NACs was also correlated with OOA from the AMS measurements.

4. Figure 8, any explanation for why the model shows poor agreement from 12/27 until 12/31?

References:

Lee, B. H., F. D. Lopez-Hilfiker, C. Mohr, T. C. Kurten, D. Worsnop, and J. A. Thornton (2014), An iodide-adduct high-resolution time-of-flight chemical-ionization mass spectrometer: Application to atmospheric inorganic and organic compounds, Environ. Sci. Tech., 48, 6309-6317.

---

## Referee Comment (RC2) · Anonymous Referee #2 · 13 Nov 2020

General. This is an interesting study on the occurence of nitroaromatic compounds (NAC) in particles which differentiates between biomass burning (BB) and secondary formation (SF) in the city of Dezhou, Shandong province in China in the winter of 2017. The paper combines NAC measurements in the gas- and particle phase with box modelling. Main measurement instrumentation was a Figaero inlet coupled to a Api-ToF-CiIMS operated in iodide cluster ion mode.

Overall, this is a very interesting study extending our knowledge on NACs significantly by state-of-the-art measurements. It is fully in the focus of ACD and, in my view, could be accepted after minor revision.

[Figure]

Details

Section 3.1 If BB periods are to be identified, was there the possibility to use levoglu­cosan measurements ?

P 12, line 19: Aqueous chemistry could be important innvolving BB compounds - what about this possibility ? Do days with the high RH give indications for contributions fm such pathways ? Maybe such consideration can be included here.
* * *

---

## Author Comment (AC1) · 18 Dec 2020

**Ambient Nitro-Aromatic Compounds – Biomass Burning versus Secondary Formation in rural China**

Christian Mark Garcia Salvador et al.

Our point-by-point responses to the reviewer's general and specific comments are presented below with the referee comments in black, our answers in red, and suggested actions to improve the manuscript in blue.

**Response to Reviewer 1**

**General:** *This paper provides measurements of nitro-aromatic compounds in rural China using FIGAERO-TOF-CIMS and provides insight into the formation mechanisms of different compounds. This paper is well-written and a nice addition on this particular topic. I recommend publication after consideration of my comments.*

**Reply:** We appreciate the careful review and constructive suggestions.

**Major Comments:**

*1. I suggest that the authors perhaps bring up occurrence of high sulfate haze events in China during winter, which has received a lot of attention, as additional motivation for better understanding more of these surprising secondary processes that dominate PM in China during winter. I think this will broaden the readership of their work*

**Reply**: We agree with the reviewer. The oxidation of $SO_2$ plays an important role in severe haze conditions in China, which leads to the enhancement of particulate matter during these events. This clearly supports the major claim of this study, which highlights the evident role of secondary chemistry during wintertime pollution events.

**Action**: The following text was modified and extended in the conclusion section.

*"This suggests that photochemical smog plays a significant role in NAC formation even during wintertime air pollution events in rural China that are dominated by primary emissions. This is consistent with the evident contribution of secondary formation in high PM episodes in North China where oxidation of other anthropogenic traces, such as $SO_2$, plays an important role during severe haze events (An et al., 2019;Huang et al., 2019)."*

*2. I suggest that the authors provide more information regarding the calibrations in the SI. Calibrating this instrument for NACs is challenging and the atmospheric community would benefit from a detailed description of how the authors performed their calibrations.*

**Reply**: Yes, we agree

**Action**: The following statements were added in table S1 to elaborate the calibrations of the NAC.

*"A post-campaign calibration of nitrophenol, nitrocatechol, and dinitrophenol was utilized to characterize the sensitivity factor of NACs. The FIGAERO filter for the collection of particle phase was doped (20-30 μL) with freshly prepared standards in methanol solvent. The filters were then desorbed in the same way as for the field sampling."*

**Specific Comments:**

**Introduction:**

1. *Page 3, Line 28 has a few grammatical issues. Remove "causes" and also "to" in between "results" and "strong"*

**Action**: Grammatical issues were addressed, and the sentence was divided and do now read:

*"BB is considered a major driver of atmospheric NAC formation (Kahnt et al., 2013;Laskin et al., 2015). Here, the combustion of coal and wood leads to thermal degradation and pyrolysis of lignins, that results in strong emission of substituted phenols including 1,2-benzenediols (catechols) and cresol/methylphenols that in turn are precursors for the formation of NACs"*

2.  *Page 4, Line 16: please also cite [Lee et al., 2014]*

**Action**: Lee et al.,2014 is now included in the main text.

***Methods:***

*1. Section 2.1: I'd be curious to see HYSPLITs and also know if any information is available regarding boundary layer height both of which I think will help interpret the data. As the authors correctly point out, the AtChem model does not factor in meteorology and adding this information will help augment the chemical interpretation of their work.*

**Reply**: A backward trajectory analysis using the NOAA HYSPLIT Model were done. However, the results did not add significant insight into explaining the chemical evolution of the investigated compounds. There was a slight difference in air mass origin between regime 1 and 2 where most of the arriving airmasses arriving in Dezhou was coming from the northwest and stayed mostly inland. During regime 1 (Dec 14-19) most of the airmasses that arrived in Dezhou came from inner Mongolia creating little time for regional secondary formation. On the other hand, airmasses during regime 2 generally had extended time in the region which could explain an aging effect on primary emission in the region before reaching Dezhou. (see Figure R1). However, we prefer not to add these speculations to the manuscript. Furthermore, Weather Research and Forecasting (WRF) calculation indicated average highest BL at ~1300 m (usually at noon, can be up to 1500), and average lowest ~800 m (usually at night, can be as low as <300m). Again, these are estimates that potentially do not completely match real condition and we after this exploration judge the usefulness of these to explain the results to be limited and rather keep the focus on the chemical indicators defining the two regimes.

[Figure]

**Action**: No action

*2. Page 5, Line 15: is this a WHO limit or a national limit for PM2.5? Please clarify.*

**Reply**: The PM$_{2.5}$ limit presented was based on the Air Quality Standards set by the European Union.

**Action**: This was clarified in the main text with following statement

*…with an aerosol mass spectrometer typically exceeded the European Air Quality allowable limit for PM$_{2.5}$ (25 µg m$^{-3}$)*

*3. Page 6, line 3, please rename to Filter Inlet for Gases and Aerosols (FIGAERO) coupled to a chemical ionization time-of-flight mass spectrometer (TOF-CIMS).*

**Action**: Done. The terms "coupled to a" was included in the description of the main instrument and the sentence now reads:

*A Filter Inlet for Gas and Aerosol (FIGAERO) coupled to a time-of-flight mass spectrometer (ToF-CIMS) was utilized to characterize the NAC content of the gas and particle phases.*

*4. Page 7, were any reference gases added to monitor drift in signal? Were any gas phase zeros performed?*

**Reply**: There were no reference gases used instead, the signal of the reagent ion (Iodide, m/z= 126.904) provided the information on the drift of the signal of the mass spectrometer (MS). The variabilities of the raw iodide signal during the field measurement were less than 10 and 20% for the gas and particle phase analysis, which indicated the minimal drift of the CIMS signal.  During the post-processing of the data, all signals from MS were normalized to the signal of the reagent ion to account for the daily variations/drifts. Gas-phase zero was performed during the post-campaign calibration of the instrument.

**Action:** The following statements that detail the drift in signal and gas phase zeroes were included in the experimental section

*The signal of the reagent ion (Iodide, m/z= 126.904) provided the information on the drift of the signal of the mass spectrometer (MS). The variabilities of the raw iodide signal during the field measurement were less than 10 and 20% for the gas and particle phase analysis, which indicated the minimal drift of the CIMS signal. During the post-processing of the data, all signals from MS were normalized to the signal of the reagent ion to account for the daily variations/drifts. Gas-phase zero was performed during the post-campaign calibration of the instrument*

5. *Page 7, Line 21: some of these compounds are light sensitive, were they used fresh for calibration?*

**Reply:** Freshly prepared calibrants were utilized during the standard compound analysis to prevent any degradation of the NACs.

**Action:** The statement in the experimental section was modified accordingly.

*"The NACs were quantified by doping the PTFE filter of the FIGAERO with known amounts of freshly prepared authentic standards"*

**Results**

1. *Figure 2: I suggest pointing out the levoglucosan peak in the spectra and other markers used to indicate BB events.*

**Reply:** We wish to include the peaks of levoglucosan and HONO in the figure, however, the signals of these two compounds were not as prominent in the graph compared to nitrocatechol and nitrophenol due to the difference in the sensitivity of the compounds to the instrument.

**Action:** No action

2. *Page 10, Line 9: your diurnal peaks in particle phase NACs are similar to those from residual wood burning seen in Gaston et al 2016.*

**Reply:** We would like to thank the reviewer for highlighting the similarity of our results with emissions from residual wood burning.

**Action:** The following statement was included in the main text

*"The diurnal profile of particle-phase NACs was comparable with the observed profile of nitrocatechol detected from residential wood smoke (Gaston et al., 2016)"*

3. *Page 17, Lines 20-22, I'd be curious to know if the secondary production of NACs was also correlated with OOA from the AMS measurements.*

**Reply:** We observed poor correlation ($r= -0.1$ to $-0.4$) for most nitro-aromatic compounds with OOA, as illustrated in the time series profiles of major NACs and OOA in figure R2. In particular, a strong enhancement of OOA was measured in the later period (Nov 28-30), when the production of NACs was the lowest. This negative relationship between NACs and OOA may indicate an efficient loss rate through partitioning to the particle phase with subsequent condensed phase reactions.

[Figure]

***Figure R2***. *Time series profile of nitrophenol, nitrocatechol, dinitrophenol, and OOA-AMS*

**Action:** No action

*4. Figure 8, any explanation for why the model shows poor agreement from 12/27 until 12/31?*

**Reply:** Indeed, the model was not able to capture the variability of the measured nitrocatechol for those days. As indicated in the previous comment, aerosol mass increased drastically between December 27 to 31 (also humidity) which would favor the partitioning of the gas phase NACs to the particle phase and potential a loss by condensed phase process/deposition.

**Action**: The following statement was included in Section 3.5 to explain the overestimation of the model between Dec 17 to 31.

*"The modelling procedure overestimated the observed concentration from Dec 27 to 31, which was accounted to elevated mass aerosol mass concentration and increased RH which would favor the partitioning of the gas phase NACs to the particle phase and a potential loss by condensed phase processes/deposition"*

**Response to Reviewer 2**

**General.** *This is an interesting study on the occurrence of nitroaromatic compounds (NAC) in particles which differentiates between biomass burning (BB) and secondary formation (SF) in the city of Dezhou, Shandong province in China in the winter of 2017. The paper combines NAC measurements in the gas- and particle phase with box modelling. Main measurement instrumentation was a Figaero inlet coupled to a Api-ToF-CiIMS operated in iodide cluster ion mode. Overall, this is a very interesting study extending our knowledge on NACs significantly by state-of-the-art measurements. It is fully in the focus of ACD and, in my view, could be accepted after minor revision.*

**Reply:** *The authors are grateful that the reviewer recognized the main relevance and highlights of our study.*

**Details**

*Section 3.1 If BB periods are to be identified, was there the possibility to use levoglucosan measurements?*

**Reply:** *The negative ion (Iodide) ToF-CIMS with FIGAERO inlet is capable of measuring levoglucosan both in gas and condensed phase. The measurement of the levoglucosan was highlighted in figure 4 where the concentration of levoglucosan was enhanced during NAC formation regime 1 when BB episodes were evident.*

**Action:** no action

**Comment:** *P 12, line 19: Aqueous chemistry could be important involving BB compounds – what about this possibility? Do days with the high RH give indications for contributions of such pathways? Maybe such consideration can be included here.*

**Reply:** *The contribution of aqueous chemistry to the formation of NAC in this study was expected to the be limited based on the weak association between relative humidity and the mixing ratio of NACs in gas and particle phase. During the measurement, the condition in Dezhou was relatively dry (Mean R.H.: 50%) with only a few days with R.H. exceeding 70%. However, we further acknowledge the importance of aqueous chemistry in the formation of nitroaromatic compounds by citing relevant studies that provided important implications of atmospheric water content. In addition, there were one exception where the RH reach 100% and a noticeable loss of NAC were observed, i.e. 27/12-31/12.*

**Action**: Additional key aqueous phase focused studies were referred to in the following statement (at P12, line 19)

*NACs may also form in aqueous phases, particularly when the atmosphere has a high liquid water content (Harrison et al., 2005b;Vidovic et al., 2020;Vidović et al., 2018).*

*A reference to aqueous phase processes where added in section 3.5 as plausible explanation for deviation between model and measurements for the period 27/12-31/12.*

*"The modelling procedure overestimated the observed concentration from Dec 27 to 31, which was accounted to elevated mass aerosol mass concentration and increased RH which would favor the partitioning of the gas phase NACs to the particle phase and a potential loss by condensed phase processes/deposition"*